# The *Drosophila melanogaster* Y-linked gene, *WDY*, is required for sperm to swim in the female reproductive tract

Yassi Hafezi [1✉], Arsen Omurzakov [1], Jolie A. Carlisle[1], Ian V. Caldas [1], Mariana F. Wolfner [1] & Andrew G. Clark [1✉]

Unique patterns of inheritance and selection on Y chromosomes have led to the evolution of specialized gene functions. We report CRISPR mutants in Drosophila of the Y-linked gene, *WDY*, which is required for male fertility. We demonstrate that the sperm tails of *WDY* mutants beat approximately half as fast as those of wild-type and that mutant sperm do not propel themselves within the male ejaculatory duct or female reproductive tract. Therefore, although mature sperm are produced by *WDY* mutant males, and are transferred to females, those sperm fail to enter the female sperm storage organs. We report genotype-dependent and regional differences in sperm motility that appear to break the correlation between sperm tail beating and propulsion. Furthermore, we identify a significant change in hydrophobicity at a residue at a putative calcium-binding site in WDY orthologs at the split between the *melanogaster* and *obscura* species groups, when *WDY* first became Y-linked. This suggests that a major functional change in *WDY* coincided with its appearance on the Y chromosome. Finally, we show that mutants for another Y-linked gene, *PRY*, also show a sperm storage defect that may explain their subfertility. Overall, we provide direct evidence for the long-held presumption that protein-coding genes on the Drosophila Y regulate sperm motility.

[1] Department of Molecular Biology and Genetics, Cornell University, Ithaca, NY 14850, USA. ✉email: yh676@cornell.edu; ac347@cornell.edu

Y chromosomes are unique in the genome of many organisms, including mammals and Drosophila, in being haploid, male-limited, repeat-rich, highly heterochromatic, and, in particular, having reduced or no recombination[1]. The resulting selective pressures on Y chromosomes cause rapid degeneration of most protein-coding genes, yet a few genes are maintained on Y chromosomes with remarkable evolutionary persistence. Such genes are maintained for extended periods under strong purifying or sometimes positive selection, repeatedly and independently acquired in different lineages, or undergo massive copy-number amplification on the Y chromosome[2–5]. These patterns of variation indicate that selection favors placing such genes into this seemingly inhospitable genomic environment. In support of this concept, there is striking similarity in both the expression patterns and functions of many Y-linked genes[6–8].

The 40 MB *Drosophila melanogaster* Y chromosome contains only 14 known protein-coding genes[9–11]. X0 flies are male yet sterile—therefore, the Y chromosome is required for male fertility but not for sex determination or viability[12]. Six genetic loci on the Y, known as the fertility factors, contribute to this fertility function. The fertility factors were defined by a series of X-ray-induced X-Y translocations[13,14] and, remarkably, half of them were discovered to be axonemal dyneins[15,16], suggesting that the *Drosophila melanogaster* Y chromosome plays a pivotal role in sperm motility. All of the Y-linked fertility factors encode sperm proteins; three of these, kl-2, 3, and 5, are among the most abundant proteins detected in sperm proteomic analyses[17]. This is consistent with their presumed role in sperm function and more specifically as major axoneme structural components[15,18,19].

*kl-1* mutant males, in contrast to all other Y-linked fertility factor mutations, produce mature and motile sperm despite being completely sterile[20]. *kl-1* sperm are transferred to the female reproductive tract (RT) following mating but cannot be recovered from the female sperm storage organs. The specific defect that prevents *kl-1* mutant sperm from entering storage or fertilizing eggs is unknown. The molecular identity of *kl-1* remained unknown until, recently, the gene *WDY* was found to be contained within the *kl-1* region[21] and required for male fertility based on RNA interference (RNAi)[22]. Other protein-coding genes or functional repetitive elements may still reside in the *kl-1* genetic region, which is estimated cytologically to span ~3% of the length of the Y chromosome[23], and it is unclear whether *WDY* mutants produce mature sperm or show a sperm storage defect. More generally, the importance of motility for sperm storage and the mechanisms that regulate sperm motility remain poorly understood in Drosophila.

Here we generated CRISPR mutants to investigate the function of *WDY*. We demonstrate that *WDY* mutant sperm display the storage defect suggested for *kl-1*. Furthermore, mutant sperm have reduced beat frequency and are unable to swim beyond the seminal vesicle. We show that mutants that we generated in another Y-linked gene, *PRY*, also have impaired sperm storage. *WDY* shows significant changes in key amino acid residues in a conserved calcium-binding domain, suggesting the functional evolution of this gene.

A high incidence of genes with predicted sperm motility functions is seen on Y chromosomes across many species, from Drosophila to great apes[7,24]. Carvalho et al.[15] hypothesized that, in species where there is a high level of sperm competition (such as *Drosophila melanogaster*), motor proteins are specifically recruited to the Y chromosome where they can evolve without constraint from male-female antagonistic selective forces. Our study provides an in-road to studying the evolutionary logic of this association.

## Results and discussion

**WDY mutants are sterile but produce mature, motile sperm.** We used CRISPR to precisely target *WDY* (Supplementary Figs. 1 and 2, Supplementary Tables 1–4). One of the major challenges of studying the Y chromosome is in propagating sterile mutations on a haploid, sex-limited chromosome. We used a crossing strategy involving compound sex chromosomes to make and stably propagate heritable mutations in *WDY* (mutant stocks consist of $\widehat{XX}$/Y,*WDY* females and $\widehat{XY}$/Y,*WDY* males, see Methods, Supplementary Fig. 3[22]). Our crossing scheme also enabled us to identify and eliminate large chromosomal truncations that are common during genetic editing of the Y chromosome, likely due to its highly repetitive nature[22]. To evaluate the phenotype of our mutants, we then removed the compound chromosomes by selective breeding to generate X/Y,*WDY* mutant males. We confirmed each phenotype with three different *WDY* alleles, F8, C104, and C3, containing deletions of 547 bp, 545 bp, and 443 bp, respectively (Supplementary Fig. 4, Supplementary Table 4). All are large deletions close to the N-terminus that disrupt the reading frame and are therefore expected to be null; all three gave the same phenotype. We compared mutants to controls that account for the genetic background (Y$^{Tomato}$) or crossing scheme (Y$^{C7}$ and Y$^{G107}$). In individual crosses to females from a wild-type strain (Canton S), control males produced progeny, while *WDY* males were sterile (Supplementary Table 5).

To investigate the cause of this sterility, we first examined the distribution of sperm in the testes using Protamine-GFP[25], which labels sperm heads. Sperm of mutants in Y-linked fertility factors kl-2, kl-3, kl-5, ks-1, and ks-2 are eliminated before this time, during the individualization stage[22,26,27]. In contrast, we observed an accumulation of *WDY* mutant sperm in the posterior-most section of the testes, where individualized sperm accumulate while sperm coiling occurs, causing that region to bulge in the mutant (Fig. 1a–f). Sperm coiling is thought to function as a quality control step during which sperm with abnormal tails are eliminated by ingestion by the terminal epithelium[28]. The accumulation of *WDY* sperm in the posterior testes may be due to their progression being stalled by this quality control mechanism or may indicate insufficient motility to exit the testes.

In the seminal vesicles, there were fewer sperm in *WDY* mutants than in the controls (Fig. 1c, d). Yet tails of sperm from both *WDY* and control males were observed to beat after we tore open the seminal vesicles (Supplementary Movie 1 and 2). Visual inspection showed no obvious differences in the movement of *WDY* versus control sperm. These observations match Kiefer's conclusion that *kl-1* mutants were sterile but produced seemingly motile sperm[20]. Our results—that *WDY* mutations are sufficient to result in sterility, yet produce sperm that are motile—make it highly likely that *WDY* is the fertility factor known in the literature as *kl-1*.

**WDY mutant sperm are transferred to females, but do not enter the storage organs.** We next tracked the movement of Protamine-labeled sperm in the RT of wild-type (Canton S) females 30 min after the start of mating (mASM). Sperm from both control and *WDY* mutant males were found in the female's uterus (bursa) (Supplementary Fig. 5), and their tails were observed to beat when dissected out of the uterus (Supplementary Movie 3, 4). In both *WDY* and control genotypes, an open or folded conformation of the uterus correlated with the presence or absence of sperm, respectively, as expected[29,30]. We conclude that motile *WDY* sperm are transferred to females and that *WDY* seminal fluid induces conformational changes in the uterus normally.

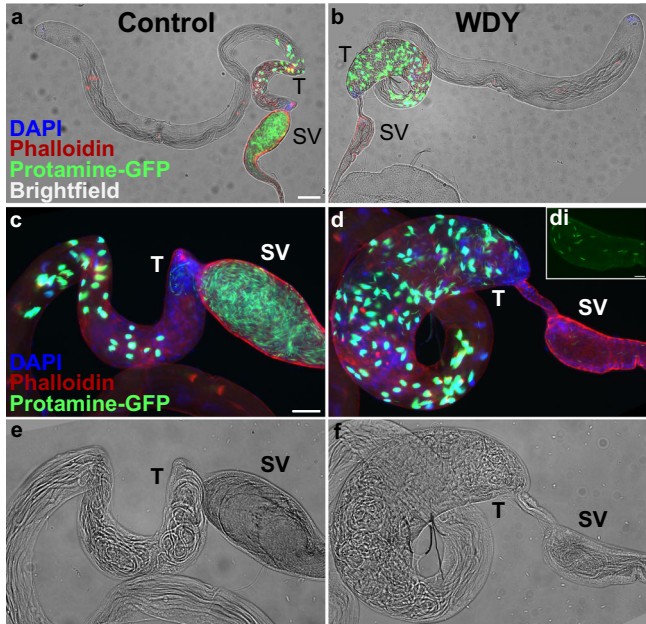

**Fig. 1 Spermatogenesis is backed up in *WDY* testes, but some mature sperm are found in the seminal vesicle.** Whole testes from control (**a**) and *WDY* mutant (**b**) males whose sperm were labeled with Protamine-GFP (green), Phalloidin (red) and DAPI (blue) overlaid on brightfield images. **c–f** Higher magnification view of posterior testes and seminal vesicle. Inset (di) shows Protamine-labeled sperm in the seminal vesicle of mutant. Testes (T) and seminal vesicle (SV) are marked. Images are representative of 6–10 samples each. Bar denotes 100 μm for **a**, **b**, 50 μm for **c–f**, and 20 μm for di.

We did, however, observe defects in the number of sperm transferred to, and the distribution of the sperm within, the female RT. *WDY* males transferred fewer than half as many sperm as control males as quantified at 30 mASM (Fig. 2a, *p* value = 8e-6, Student's *t* test). After mating, Drosophila sperm move rapidly from the uterus into either the primary storage organ, the seminal receptacle, or one of the two long-term storage organs, the spermathecae[31] (Fig. 2b). At 30 mASM most control samples contained some stored sperm, but no *WDY* sperm were found in the storage organs. At 2 hASM maximal numbers of sperm are stored in most control samples[32]. Yet, again, no sperm from *WDY* mutants were seen in storage (Fig. 2c–e). We also examined sperm in RTs of females left to mate overnight to see if a longer time or multiple mating could enable *WDY* mutant sperm to enter storage (Supplementary Fig. 5). Control samples all had stored sperm. *WDY* sperm were regularly observed in the uterus but never in any of the storage organs (Fig. 2c). We conclude that *WDY* mutant sperm are unable to enter the storage organs. In many animals, storage is required for sperm to become competent for fertilization[33,34]. Thus the lack of sperm storage might explain why *WDY* males are sterile.

**WDY mutant sperm in the male seminal vesicle and female uterus have decreased tail-beat frequency.** Although *WDY* mutant sperm beat visibly in vitro (Supplementary Movie 1–4), we wished to test whether subtle motility defects prevent them from being able to enter storage. We measured the beat frequency of sperm tails, by recording videos of control and *WDY* mutant sperm dissected directly from the male's seminal vesicle or from the female's uterus at 30 mASM (Methods, Supplementary Movie 1–4, Fig. 3a, b). In the seminal vesicle, the fastest *WDY* mutant sperm tails beat at an average of 6.0 Hz, whereas the

fastest control sperm tails beat at an average of 12.3 Hz (p-value = 3.5e-10, Likelihood Ratio Test). In the uterus, the fastest *WDY* mutant sperm tails beat at an average of 7.0 Hz, whereas the fastest control sperm tails beat at an average of 13.1 Hz (*p* value = 7.3e-6, Likelihood Ratio Test). We conclude that *WDY* mutant sperm have a lower tail-beat frequency than wild-type sperm in both the male and female RTs.

**WDY mutant sperm are unable to swim in the male ejaculatory duct and female uterus.** We hypothesized that the lower tail-beat frequency affects the ability of *WDY* mutant sperm to propel themselves. To test for defects in sperm swimming, sperm movement was assessed in videos by tracking the protamine-labeled heads of control and *WDY* mutant sperm. In all regions the swimming speed of individual sperm varied, but there was an overriding regional pattern to the motility (Fig. 3c–j). In mammals, sperm leaving the testes are immotile and must go undergo epididymal maturation in order to gain the ability to move progressively[35]. It was previously suggested, and has often been repeated, that, similarly, Drosophila sperm do not gain motility until they reach the seminal vesicle[36]. We were thus surprised to see some individualized sperm heads slowly swimming within the posterior testes of most control samples (Fig. 3c, ci, Supplementary Movie 5). This suggests that, in Drosophila, sperm motility is normally initiated within the testes. *WDY* mutant sperm heads in this region also often moved around, suggesting that at least some mutant sperm develop motility (Fig. 3d, di, Supplementary Movie 6).

Individual sperm heads generally ceased to move in the seminal vesicles of both control and *WDY* mutant flies, while flagella remained beating. However, mass movements occurred from contractions of the whole organ. It was unclear whether dense packing of sperm or some physical or chemical property of the seminal vesicle caused the immobilization of sperm heads while sperm tails continued to beat vigorously (Fig. 3e, ei, Supplementary Movie 7). This highlights that flagellar beating does not necessarily correlate with sperm swimming (i.e., moving through space). There were far fewer sperm in *WDY* mutant seminal vesicles, but the mutant sperm heads were predominantly immobilized, as in controls (Fig. 3f, fi, Supplementary Movie 8). We conclude that at least some *WDY* sperm develop the ability to swim in the testes and become immobilized in the seminal vesicle, as normal.

In contrast, a striking difference was seen in the ability of *WDY* sperm to swim beyond the seminal vesicle. In samples where sperm were found in the ejaculatory duct, control sperm heads were observed to move swiftly while *WDY* sperm heads appeared motionless (Fig. 3g, h, gi–hi, Supplementary Movie 9, 10), and the same pattern was observed for sperm heads in the uterus 1 hASM (Fig. 3I, j, ii, ji, Supplementary Movie 11, 12). These sperm appear to be alive, as their tails continue to beat in place (Fig. 3a, b). The inability of *WDY* sperm to enter the sperm storage organs likely reflects their diminished swimming in the uterus, though other defects may also exist and contribute[37]. That *WDY* sperm in the posterior testes can swim suggests either (1) there is a subclass of *WDY* sperm that are capable of swimming but degenerate, or (2) *WDY* mutant sperm are unable to navigate between different regions of the RT.

**Significant hydrophobicity differences in putative calcium-binding residues coincide with WDY's transition to Y-linkage in the melanogaster lineage.** Calcium regulates sperm motility in many organisms, including humans[38] and Drosophila[39–41]. WDY's amino acid sequence contains a calcium-binding domain signature: an EF Hand (Interpro[42]). Functional EF Hand domains

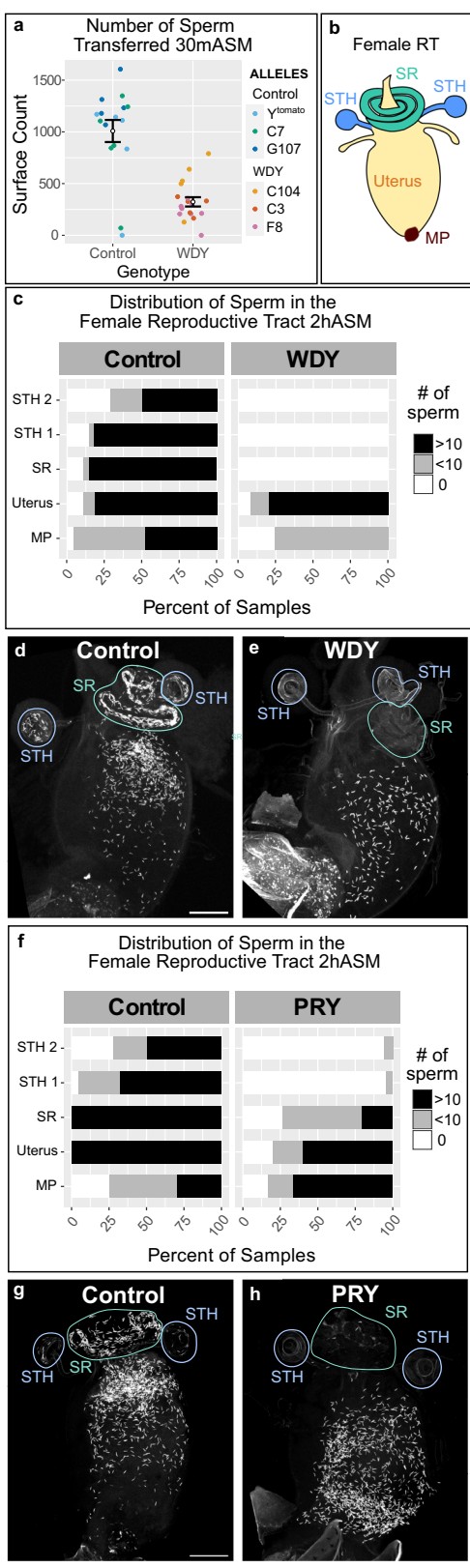

**Fig. 2 *WDY* and *PRY* mutant sperm fail to enter the storage organs in the female RT. a** Quantification of the number of sperm transferred to the female uterus 30 mASM by control ($n = 16$ reproductive tracts) or *WDY* mutant males ($n = 18$ reproductive tracts). Bars indicate standard error; $p$ value = 8e-6 by Student's $t$ test. **b** Cartoon of the female RT indicating the mating plug (MP, brown), uterus (yellow), and the storage organs—the seminal receptacle (SR, green) and two spermathecae (STH, blue). **c–h** Quantification of the distribution of Protamine-labeled sperm within the female RT 2hASM. **c** Distributions of sperm from control males ($n = 28$ female RTs) and *WDY* mutant males ($n = 25$ female RTs) were compared ($p$ value < 2.2e-16, Asymptotic Linear-by-Linear Association Test). Representative images shown in **d**, **e**. **f** Distributions of sperm from control males ($n = 25$ female RTs) and *PRY* mutant males ($n = 20$ female RTs) were compared ($p$ value = 5.6e-15, Asymptotic Linear-by-Linear Association Test). Representative images shown in **g**, **h**. Bars denote 100 μm.

display this configuration. We also improved the annotation of two WD40 domains (Methods, Fig. 4a, Supplementary Table 6), which typically mediate protein-protein interactions in protein complex assembly and/or signal transduction. Based on these findings, we speculate that *WDY* is necessary for sperm to adjust their motility based on differences in calcium levels in different regions of the RT.

We also compared sequences of the EF Hand domain between WDY orthologs in Drosophila species from three groups encompassing: (1) the initial transition of WDY to the Y chromosome from its ancestral autosomal site during the *melanogaster-obscura* group split, (2) a whole-chromosome Y-incorporation event in the montium subgroup, in which the Y chromosome is thought to have become duplicated elsewhere in the genome, and (3) the subsequent reestablishment of Y-linkage in the *kikkawai* clade (Fig. 4b, c,[3,4]). We note a higher rate of sequence divergence of the Odd EF Hand Motif (55.9% Identity, Fig. 4b) relative to that of the Even EF Hand motif (92.7% Identity, Fig. 4c) in Drosophila sequences. Similarly, we observed a 12-fold difference in $d_N/d_S$ between the Odd EF hand (omega = 0.00239) and the Even EF hand (omega = 0.00020) in a pairwise comparison between *D. melanogaster* and *D. obscura* WDY using Model 0 of codeml. This supports the idea that the Odd EF hand sequence is diverging more rapidly or has more relaxed purifying selection than the Even EF hand sequence.

The amino acid transition in the Odd EF Hand Motif at position "X" is particularly compelling, since it involves a profound biochemical change in a conserved residue thought to directly bind calcium[44] (Fig. 4d, e). The shift away from canonical residues in the *melanogaster* group could indicate a modulation of calcium binding, and, thus significant functional evolution in the EF Hand domain, coinciding with WDY's initial Y-linkage. Corresponding shifts in hydrophobicity are not observed to be correlated with the genomic movements of WDY in the *montium* subgroup, but we would not necessarily expect a change that occurred when a gene moved to the Y chromosome to reverse if the gene moves off the Y chromosome. Moreover, selective pressures that drove a chromosome-wide Y-incorporation event are likely to have been significantly different from those driving movement of a single gene onto the Y chromosome. Future functional studies will be required to formally test the significance of the change in hydrophobicity at position "X".

A previous publication from our lab identified signatures of positive selection in WDY. We hypothesized that positively selected sites may be present within the EF-hand domain[2]. We thus updated the analysis of variation in WDY by including

contain a pair of motifs, each consisting of a loop flanked by alpha helices, that can bind $Ca^{2+}$ ions. The specific characteristics of the loop affect calcium-binding affinity[43]. We identified a putative pseudo EF hand motif followed by a canonical EF hand motif in WDY (Methods, Fig. 4a–c, Supplementary Table 6). Known calcium-binding proteins (e.g. Calbindin D9K[44]) also

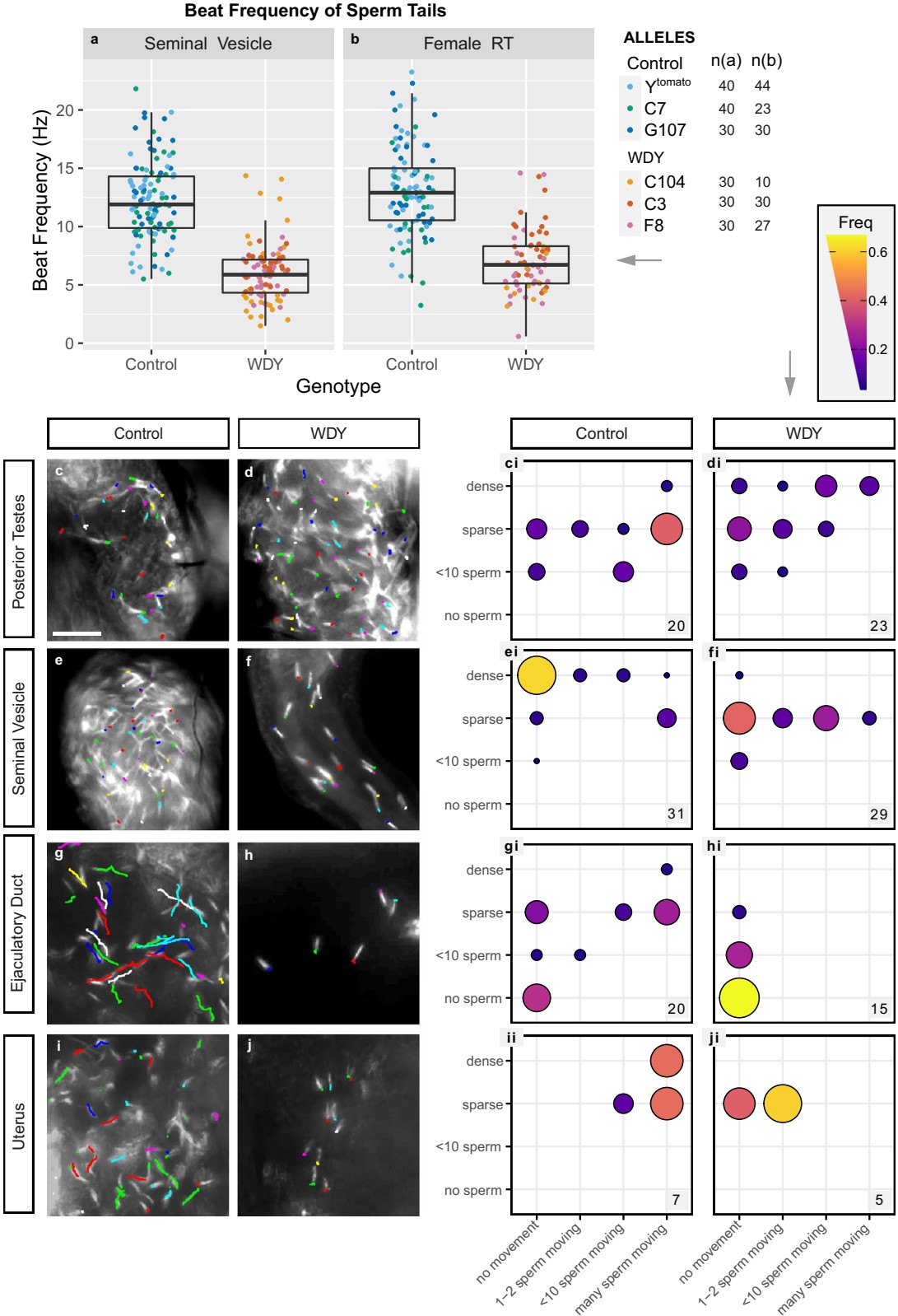

sequences from thirteen addition species (Supplementary Table 7), a more stringent model comparison and statistical analysis to account for neutral evolution, and running the analysis on individual Drosophila clades, to avoid synonymous saturation. Our updated analysis (Supplementary Tables 8 and 9) indicates that there is no evidence of positive selection that could not be better explained by neutral evolution. Furthermore, posterior probabilities for site classes, determined by Bayes Empirical Bayes ($p > 0.9$), identified no specific sites under positive selection. Similar results were obtained with a branch-site test on full-length WDY (Supplementary Table 10). Therefore, although there seems to be an amino acid change in the EF hand domain as WDY became Y-linked, these sites cannot be shown to be undergoing positive selection. The initial movement of a gene to the Y

**Fig. 3 WDY mutant sperm have reduced tail-beat frequency and do not swim in the female RT. a**, **b** Quantification of tail-beat frequency of sperm. Boxplots show median, first and third quartiles, and values within 1.5× the interquartile range for each genotype. Numbers quantified are indicated next to the legend (n = sperm tails per genotype). **a** WDY and control sperm dissected from seminal vesicles (p value = 3.5e-10, Likelihood Ratio Test) and **b** WDY and control sperm dissected from the female RT 30 mASM (p value = 7.3e-6, Likelihood Ratio Test). **c–ji** Manual tracking of sperm heads. Protamine-GFP-labeled control (**c**, **e**, **g**, **i**) and WDY mutant (**d**, **f**, **h**, **j**) sperm heads over a 0.5 sec interval for representative videos of the posterior testes (**c**, **d**), seminal vesicle (**e**, **f**), ejaculatory duct (**g**, **h**), and uterus 1 h ASM (**I**, **j**). (**ci–ji**) Corresponding quantification of the number of sperm heads and degree of movement observed from videos of each region of the RTs. The number of each type of organ that was scored is indicated on each plot. Bar denotes 25 μm for **c–j**.

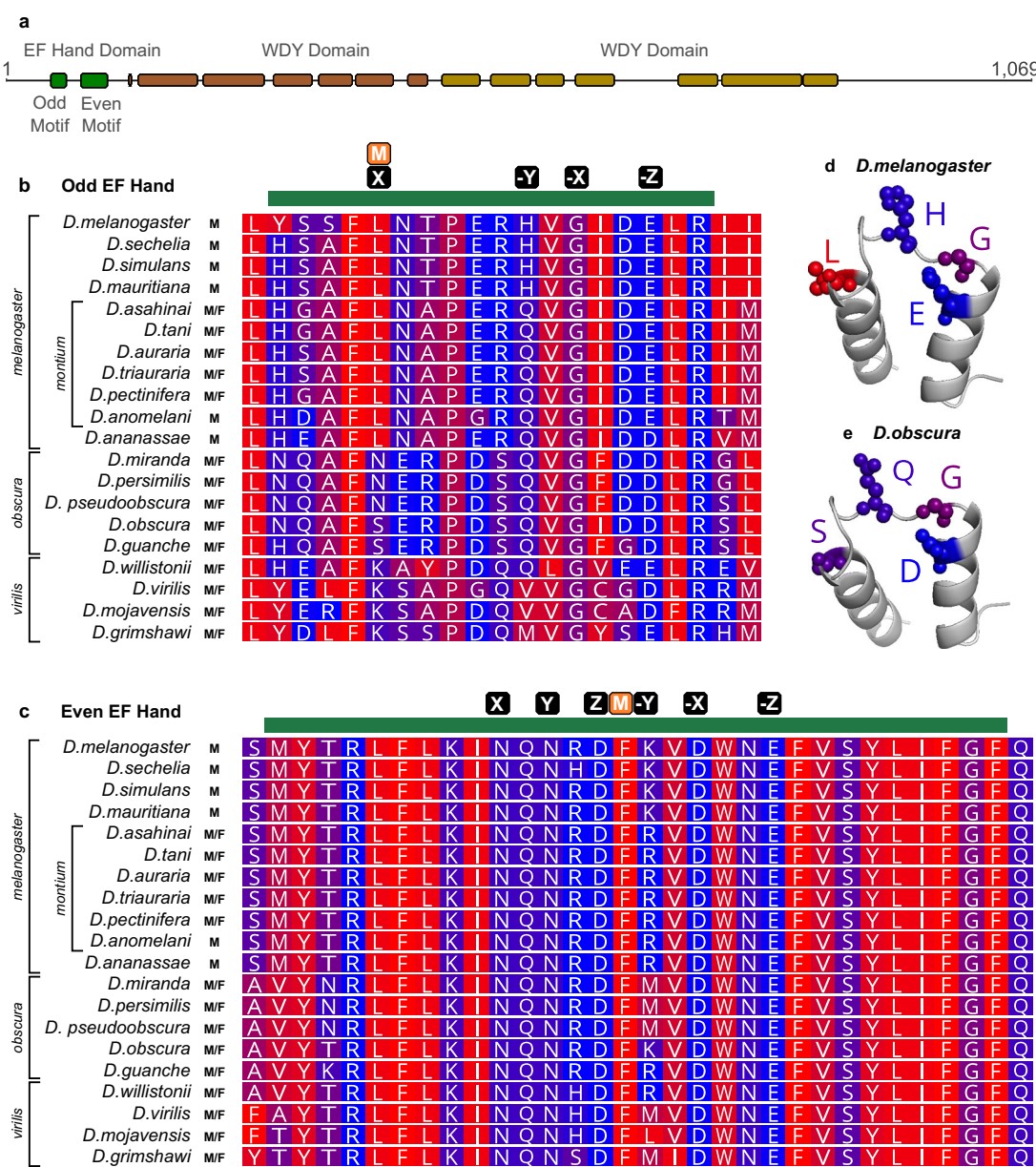

**Fig. 4 Difference in hydrophobicity in the Odd EF hand domain motif coincide with the initial movement of WDY to the Y chromosome at the melanogaster-obscura species group split. a** Domain structure of WDY. **b**, **c** Protein alignment of melanogaster and obscura group species for the region with the Odd (Pseudo) (**b**) and Even (Canonical) (**c**) EF Hand motif. Green bar indicates the motif, M indicates the position of any mismatch between the melanogaster sequence and the consensus, and black boxes indicate putative calcium-binding residues (X, Y, Z,-X,-Y,-Z). Blue-red scale indicates hydrophobicity (Red is hydrophobic, Blue is hydrophilic). **d**, **e** Predicted AlphaFold structure of Odd EF Hand domain with putative calcium-binding residues labeled, generated in PyMOL for Drosophila melanogaster (**d**) and Drosophila obscura (**e**).

chromosome may be more significant for its functional evolution than subsequent movements onto/off of the Y chromosome. This is consistent with the observation of high incidence of gene reappearance on the Y chromosome after Y-incorporation events[4].

**PRY is also required for efficient sperm storage**. We previously generated and characterized mutants in another Y-linked gene, PRY, whose phenotype was consistent with abnormal sperm storage in females: mutants had low levels of fertility on the first

day after mating but no fertility on subsequent days[22]. Our finding that *WDY* affects sperm entry to the storage organs led us to wonder whether *PRY* affects a similar step. Indeed, the number of *PRY* sperm stored was significantly reduced compared to controls at 2 hASM (Fig. 2f–h, *p* value = 5.6e-15, Asymptotic Linear-by-Linear Association Test for difference in sperm distribution). *PRY* mutant sperm were frequently absent or reduced in the seminal receptacle, and rarely observed in the spermathecae. The number of stored *PRY* sperm remained low at 24 hASM (Supplementary Fig. 5, *p* value < 2.2e-16, Asymptotic Linear-by-Linear Association Test for difference in sperm distribution), however, significantly more sperm entered storage organs if males and females were housed together overnight (Supplementary Fig. 5, *p* value = 1.9e-5 for comparison of *PRY* versus control after overnight mating, 5.1e-6 for comparison of *PRY* 24 hASM versus after overnight mating, and 0.9462 for comparison of control 24 hASM versus after overnight mating, Asymptotic Linear-by-Linear Association Tests for difference in sperm distributions). Seminal fluid proteins responsible for long-term physiological effects of mating on females, including the inhibition of remating, can bind to sperm tails[2,45–47]. Thus, lack of stored *PRY* sperm may lead to increased remating in these females.

In contrast to *WDY*, *PRY* mutant sperm did swim in the female RT (Supplementary Movie 13) and the beat frequency of the fastest-beating *PRY* mutant sperm tails (12.2 Hz) was not significantly different from the corresponding control's (11.5 Hz, Supplementary Fig. 6, *p* value = 0.5316, Likelihood Ratio Test). Defects in other aspects of motility or navigation[48] may prevent the majority of *PRY* mutant sperm from entering sperm storage.

## Conclusions
Overall, we present functional evidence for a role for two Y-linked genes in sperm storage and demonstrate that *WDY* mutants have specific defects in sperm motility. Our work extends Kiefer's intriguing observation that *kl-1* mutants produce beating sperm despite being sterile. *WDY* mutants recapitulate this unusual phenotype. In the absence of the ability to test complementation, this strongly suggests that *kl-1* and *WDY* mutants perturb the same gene.

Two broad themes emerge from this work. First, sperm may be unable to enter storage for different reasons—insufficient swimming speeds or an inability to navigate or gain entry into the storage organs. We demonstrate that sperm motility defects do manifest in an inability to enter storage. Female secretions are necessary to promote sperm storage[49], so entry to sperm storage may be a hurdle imposed by females to ensure that only sperm with a certain level of motility/fitness are able to fertilize eggs.

Second, across species, Y-linked genes appear to show 'functional coherence'[6,26]. Even within the realm of male fertility, a disproportionate number of Y-linked genes seem to be singularly focused on aspects of sperm motility[15,16]. Three axonemal dyneins were previously discovered on the Drosophila Y chromosome[15,16]. However, since no sperm are produced upon genetic ablation of five out of the six fertility factor genes, the role of these genes in sperm motility was never previously tested[13,14,22,26]. We now add strength and stringency to the picture of functional coherence on the Drosophila Y chromosome. On the one hand, being Y-linked allows sperm motility genes to escape the problems of countervailing selection in females (sexual conflict). However, being Y-linked bears the cost of not being able to recombine, which reduces the efficacy of natural selection (the Hill-Robertson effect[1]). The fact that so many sperm motility genes are retained on the Y chromosome indicates a dynamic balance between these two opposing selective forces in different regions of the genome.

## Methods
**Drosophila stocks and husbandry**. Flies were reared on a cornmeal-agar-sucrose medium (recipe available at https://cornellfly.wordpress.com/s-food/) at 25°, with a 12 hr light–dark cycle. The stocks used in this study are described in Supplementary Table 1.

**Generation of a *WDY* Mutant with CRISPR**. Three 20-base pair guide RNAs were designed to target exon 2 of *WDY*, a region of the gene with no known duplications (Supplementary Fig. 1)[11]. We also targeted *ebony*, a visible Co-CRISPR marker[50]. Guide sequences were incorporated into pAC-U63-tgRNA-Rev (Addgene, Plasmid #112811), which is analogous to the "tgFE" construct from[51]. This was done by appending guide RNA sequences to tracrRNA core and tRNA sequences from pMGC (Addgene, Plasmid #112812) through tailed primers (Supplementary Table 3) to create inserts that were then inserted by Gibson Assembly into a SapI-digested pAC-U63-tgRNA-Rev (Addgene, Plasmid #112811). The plasmid backbone contained attB, and we used PhiC31 to integrate it into an attP-9A site on chromosome 3 R. The construct was injected into *yw nanos-phiC31; PBac{y + -attP-9A}VK00027* by Rainbow Transgenic Flies Inc. Transformants were identified by eye colour from a *mini-white+* marker. Transformants express the four guides ubiquitously under the U6:3 promoter as a single polycistronic transcript that is processed by the endogenous cellular tRNA processing machinery (RNase P and Z) to release the individual mature gRNAs and interspersed tRNAs. The transformants were balanced, and inserts were confirmed by PCR and sequencing. A few of the transformants had light-red eyes, but we only used those with dark-red eyes.

The combination of transformants and germline Cas9 drivers was optimized for efficiency. Males containing *vasa-Cas9* and our guide RNAs were sterile, while females showed a 6.2% CRISPR efficiency based on the generation of *ebony* mutants. In contrast, F1 males from crosses with *nanos-Cas9* drivers on chromosomes 2 and 3 produced progeny. F2 progeny (from male and female crosses combined) showed that these two lines had editing efficiencies of 2.2% and 2.6%, respectively. Our observation of higher efficiency and sterility from *vasa-Cas9* is consistent with the earlier[52] and higher somatic[53,54] protein expression of Vasa versus Nanos, as well as the RNAi phenotype of *WDY*[22]. We proceeded to make stable mutants by crossing transformant #1 to *nanos-Cas9* in a compound chromosome background.

Our crossing scheme for creating *WDY* alleles is shown in Supplementary Fig. 3. We combined the *nanos-Cas9* driver on chromosome 3 with a Y chromosome marked with *3xP3-tdTomato*[55]. We also combined the guide-expressing insert on 3 R with a compound X ($\widehat{XX}$; *C(1)M4, y[1]*). CRISPR editing occurred in F1 females ($\widehat{XX}/Y^{3XP3\text{-}tdTomato}$) that carried the marked Y chromosome. By crossing to a compound X-Y ($\widehat{XY}$, *C(1;Y)1, y[1]*) we were able to establish balanced lines from 55 *ebony* and 5 non-*ebony* F2 flies. Males were of genotype $\widehat{XY}/Y^{3XP3\text{-}tdTomato}$ and were fertile regardless of CRISPR-mediated edits of the free Y chromosome. We screened these lines for visible deletions in the *WDY* target site—first by gel, then by sequencing. Alleles derived from our crossing scheme are listed in Supplementary Table 4 and described in Supplementary Fig. 4. They are maintained as stable lines with $\widehat{XX}/Y$ females and $\widehat{XY}/Y$ males; the free Y chromosome is edited.

In several of our lines, we saw varying, intermediate degrees of position effect variegation (PEV) (Supplementary Fig. 2). This corresponded with either failed amplification at the target site in *WDY* or the presence of several bands of unexpected size. Based on our previous results when editing *FDY* with CRISPR[22], we hypothesized that these mutations were large deletions in the Y chromosome Thus, we did not phenotype these mutants for sperm or fertility characteristics. C(1)M4 contains *white^mottled-4^*, a PEV marker that is highly sensitive to Y chromosome dosage. $\widehat{XX}$ females with C(1)M4 have mostly white eyes, while $\widehat{XX}$ females have an almost entirely red eye. We previously showed that lines with visibly altered PEV lacked large sections of the Y chromosome[22]. Such deletions may be caused by the presence of uncharacterized copies of the target region present in unassembled regions of the Y chromosome.

**Sterility, mating, and sperm storage**. Crosses and experiments were done with flies 2–5 days after eclosion (dAE). To test for sterility, we crossed individual XY males to 4 Canton S virgin females in a food-containing vial with wet yeast. Adults were transferred to a new vial after one week. Crosses were scored for the presence of progeny. 15–20 crosses were tested per line. For experiments that required timing from the start of mating, one Canton S virgin female was mated to three males of a given genotype, and flies were observed. Once mating began the time was noted. Females were analyzed or flash frozen in liquid nitrogen 30 minutes, 2 hours, or 24 hours after the start of mating (30 mASM, 2 hASM, 24 hASM). Reproductive tracts were dissected from frozen females in PBS, fixed in 4% paraformaldehyde, and mounted in Vectashield with DAPI. Samples were imaged on an Echo Revolve microscope or a Leica DMRE confocal microscope. The distribution of sperm in the female reproductive tract was assessed from the images. Each region was scored as containing 0, 10 or fewer, or >10 sperm.

**Sperm counting with Imaris software**. To quantify sperm transferred, female reproductive tracts 30 mASM were imaged on a Leica DMRE confocal using standardized settings. Two μm Z-stacks through each sample were collected. Using Imaris 9.8.0 software (RRID:SCR_007370), first the female reproductive tract was extracted in each image by manually drawing a contour surface. The mating plug and cuticle were specifically excluded due to their high autofluorescence[56]. Second, protamine-labeled sperm heads were automatically detected using the "Surfaces" function (smoothing and background elimination enabled, 2.0 μm surface grain size, 1 μm diameter of largest sphere, 2.747–13.048 manual threshold, >15 quality, <0.8 sphericity).

**Sperm tail-beat frequency analysis**. Tail-beat frequency was measured for sperm dissected from the reproductive tracts of females 2–5 dAE and 30 mASM or males 2–5 dAE into PBS. Sperm were released into a 15 μl drop of PBS on a glass slide by tearing the male seminal vesicle or female uterus. Sperm were observed under brightfield optics with an Olympus BX51WDI microscope and a ×50 LMPLFLN objective. Eight-second raw movie clips at $1280 \times 720$ resolution and 60 frames per second were captured from 4–6 different regions around the sperm mass using a Canon EOS Rebel T6 camera. Dissected sperm masses all contain sperm tails beating at a range of frequencies. We specifically quantified the beat frequency of the 1–2 fastest-beating sperm tails from each clip.

To measure sperm tail-beat frequency, video clips were imported to FIJI (RRID:SCR_002285) using the ffmpeg plugin. From each clip, we measured beat frequencies of the 1-2 fastest-beating sperm and limited to tails that were not overlapping or entangled with other sperm tails. A selection line was drawn across an isolated section of sperm tail. A 1-pixel "Multi Kymograph" was generated, which shows pixel intensities across the selection line on the *X* axis for each frame along the Y-axis. The beating of the sperm tail appears as a traveling wave form. The number of beats and the number of frames were counted for the region where the sperm tail remained in focus and isolated from other tails. Beat frequency was then calculated as: Hz = (no. of beats $\times$ 60 fps)/no. of frames.

**Sperm swimming analysis**. Videos of sperm swimming were acquired from either male reproductive tracts or female reproductive tracts 1 hASM. Tracts were dissected and mounted in 15 μL PBS. Spacers (2 layers of double-stick tape) were used to avoid compression of the tissue by the coverslip. Fluorescent sperm heads were recorded through screen recording of the preview window on an Echo Revolve. We used ffmpeg (RRID:SCR_016075) to convert videos to constant frame rate of 60 fps and. mov format. Videos were then imported into FIJI (RRID:SCR_002285) using the ffmpeg plugin. We manually tracked sperm heads across 60 frames using the "Manual Tracking" plugin in FIJI. The tracking shown in Fig. 3 represents movement across 30 frames.

**Annotation of *D. melanogaster* WDY**. EF hand motifs were identified by searching (using Geneious software, RRID:SCR_010519) for the canonical and pseudo PROSITE motif consensus sequences defined in Zhou et al.[44] and allowing for a maximum 1 base pair mismatch. Because the pseudo EF hand motif contains a variable size region, there were two potential start locations in the sequence—residue 44 or 47. However, Alphafold prediction showed residues that should form the loop region would instead form part of the alpha helix in the motif beginning at residue 44. We, therefore, favored the motif beginning at residue 47. Locations of the calcium-binding residues were determined based on the consensus sequence logograms[44].

Three WD40 domains[21] were originally identified in the protein sequence based on homology. Flybase reported a handful of WD40 repeats (2 for Pfam and 8 for SMART) were identified. 4–16 of these repeat domains may together form a circular beta-propeller structure called a WD40 domain[57–59]; however, insufficient WD40 repeats were identified in WDY to predict the presence of a WD40 domain. We used the structural prediction of *D. melanogaster* WDY by Alphafold (PDB B4F7L9)[60] to identify the locations of the characteristic β-propeller, consisting of four antiparallel sheets[58]. WDY is predicted to form two WD40 domains–one with 6 WD40 repeats and one with 7 WD40 repeats.

**Identification of WDY ortholog sequences**. WDY ortholog sequences were obtained from GenBank, Chang et al.[61], or extracted from publicly available genomes using Exonerate version 2.2.0[62], as noted in Supplementary Supplementary Table 7. Newly extracted sequences were obtained by aligning the *D. melanogaster* WDY protein sequence (NM_001316659.1) to the genomic scaffolds containing WDY in other species via the Protein2Genome command. The top-scoring prediction from Exonerate was used to define the sequence.

For the comparisons in Fig. 4, protein sequences were aligned in Geneious software (RRID:SCR_010519) using a BLOSUM cost matrix with a gap open cost of 10 and a gap extend cost of 0.1.

**Selection analysis**. To create sequence alignments for selection analysis, we translated the WDY coding sequences and then aligned the protein sequences with MAFFT[63]. Protein alignments

were converted to nucleotide alignments by PAL2NAL[64]. Sites, where >50% of the species had a gap in the alignment were removed from the final alignment used in the analysis. We pruned the phylogeny published in Kim et al.[65] to include only relevant species as input phylogenies for codeml analysis.

We used the codeml program of PAML 4.8[66] to determine if there was evidence of positive selection in WDY and to potentially identify specific codons subject to positive selection. We compared neutral models to models including positive selection (M1a v. M2a, M7 v. M8, and M8a v. M8) via a likelihood ratio test (LRT). The LRT statistic was calculated from the model likelihoods as $2*[\log(L_a) - \log(L_0)]$, where $L_a$ and $L_0$ are the likelihoods under the alternate and null hypotheses, respectively. For M1a v M2a and M7 v M8 comparisons, the LRT statistic was compared to the Chi-squared distribution with 2 degrees of freedom[66]. For the M8a v M8 comparison, 1 degree of freedom was used[67]. Specific codons evolving under positive selection were identified via M8 in codeml with Bayes Empirical Bayes probabilities >0.9[67]. We also used model 0 in codeml to obtain $d_N/d_S$ values on a pairwise comparison between *D. melanogaster* and *D. obscura* and performed a branch-site model test (M0 v. M2) using all available WDY sequences.

**Statistics and reproducibility**. Statistical analysis was done using R and R-Studio (RRID:SCR_001905).

Distributions of sperm in the female reproductive tract were examined for 20–30 samples for each genotype. Differences in the distribution of sperm between different genotypes were compared using an Asymptotic Linear-by-Linear Association Test for an ordered categorical variable by genotype, stratified by reproductive tract region.

Counts of transferred sperm from control and *WDY* males showed homogeneity of variance and were approximately normal. They were therefore statistically compared using a Student's *t*-test.

Sperm tail-beat frequencies were measured from a minimum of three individuals for each allele and ten measurements per individual. Using the lme4 package in R, linear mixed models were fitted to the data, incorporating the individual as a random effect and the experimental batch and the experimenter who measured beat frequency as fixed effects. We then ran a Likelihood Ratio test to compare the model with and without "Genotype" as a fixed effect. Approximately one-third of samples were scored blind, and statistical analysis indicated consistent results whether or not samples were scored blind.

**Reporting summary**. Further information on research design is available in the Nature Portfolio Reporting Summary linked to this article.

## Data availability
Data from this study, including the raw images, data tables, and videos from the phenotypic analysis, *D. melanogaster* WDY protein annotation file, and selection analysis files (alignments, phylogenies, control files, and output files), are publicly available through Cornell eCommons: https://doi.org/10.7298/vtr8-ab60.

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

## Acknowledgements

We thank Yaoyi Xing, Emeka Okorie, Julia Kelso, Elissa Cosgrove, Nora Brown, Sarah Allen, the Imaging Facility at the Cornell Institute of Biotechnology (RRID:SCR_021741), the Cornell Statistical Consulting Unit and members of the Clark and Wolfner laboratories for advice and assistance with experiments or analysis. We thank Susan Suarez and the reviewers for their feedback and advice on the experiments and manuscript. Stocks obtained from the Bloomington Drosophila Stock Center (National Institutes of Health grant P40OD018537) were used in this study. This work was supported by funds from the National Institutes of Health (R01-HD059060 to A. Clark and M. Wolfner and R01-GM119125 to A. Clark and D. Barbash) and a seed grant from the Cornell Center for Vertebrate Genomics.

## Author contributions

Y.H. and A.G.C. conceived the study. Y.H., J.A.C., M.F.W., and A.G.C. designed the experiments. Y.H. and A.O. made the *WDY* CRISPR mutants. Y.H. performed the phenotypic experiments in Figs. 1–3. I.V.C assisted with the image analysis in Fig. 3. Y.H. and J.A.C. performed the orthologue comparisons in Fig. 4. J.A.C. performed the selection analysis and annotation of WDY. Y.H., J.A.C., M.F.W., and A.G.C. wrote the manuscript.

## Competing interests

The authors declare no competing interests.
