## [Peer Review File · Communications Biology]

Reviewers' comments:

Reviewer #1 (Remarks to the Author):

This manuscript focuses primarily on the functional role of a Y-linked gene, WDY, in the fertility of the *D. melanogaster* males. For that, the authors generate CRISPR mutants, finding evidence of decreased sperm motility and unusual distribution patterns in the female sperm storage organs relative to the wild type. The authors characterize different properties of the mutant sperm in order to narrow-down the effects on male fertility. Specifically, they find that WDY mutant sperm has low tail-beat frequency - compared to wildtype sperm- and that cannot swim in the uterus, preventing the sperm from entering into the spermathecas and the seminal receptacle. A similar, but more limited, analysis is done on the mutant for the PRY gene, which is also Y-linked. The results are also interesting, finding important differences with the conclusions related to the WDY mutant. Overall, mutant characterization is comprehensive and convincing. The results help us to better understand the genotype-phenotype link in the context of the functionality of Y-linked genes. The dissociation between tail beating and actual propulsion is an additional relevant observation.

Specific comments and concerns:

1. The section on the hydrophobicity differences in putative calcium-binding residues and the transition to Y-linkage in the *Drosophila melanogaster* species group is very different from the rest of paper, more speculative, and the implications based on correlative patterns. It would be nice to see a better connection between their observations and previous results identifying the action of positive selection (Singh et al. 2014) or relative to new, more refined analyses that can identify particular codons as target of positive selection. Do those codons coincide with the loop residues of the Odd EF Hand motif?
2. In the Conclusions section, I miss some discussion on the relationship between WDY and kl-1.
3. Please indicate in the main text the statistical test used next to the p-value.

Reviewer #2 (Remarks to the Author):

Hafezi et al. have produced CRISPR KO of two Y-linked genes in *Drosophila* and conducted a very thorough functional analysis of the resultant sperm and sperm storage phenotypes. The function and evolution of Y-linked genes has been of great interest to the field with regard to spermatogenesis and sexual selection but advances have been limited given the challenges of doing proper genetics involving the Y because KOs are going to result in male infertility. The authors have used a very clever crossing strategy using compound sex chromosomes to allow them to generate males with WDY and PRY KOs. So the combined use of CRISPR KOs and this mating strategy is very timely and provides the first real opportunity to carefully interrogate the functional role of Y-linked fertility factor. The analyses involved assay of sperm storage and in vitro sperm motility and were informative and robust. The manuscript is also very well written. As such, I only have a small number of substantive comments that need to be addressed. In summary, a very well-designed and informative study about how Y-linked genes contribute to sperm function and fertility.

Specific Comments:

1. Page 5; 3rd paragraph: WDY molecular evolution- Phylogeny almost certainly explains some, if not all, of this pattern between groups. A proper molecular evolutionary analysis is required to account for divergence between the *melanogaster* and *obscura* group. Furthermore, isn't the expectation that WDY should evolve more rapidly once it moves to the Y? This could also be part of the model that is tested. Is the radical change at amino acid residue "X" a selected site? A formal analysis of WDY

evolution should be conducted to better understand the selective dynamics associated with its relocation to the Y.

2. Page 6; 2nd paragraph; last sentence: I'm a bit uncomfortable with this sentence even though the statements are qualified. First, no analyses of remating rates were conducted. It seems a bit premature to connect the dots here between reduced sperm storage numbers, SFP abundance in the FRT and remating rates. Furthermore, it also has the potential to be misleading to connect this to sperm competition, in turn. Several *Drosophila* genes now have "sperm competition" GO characterizations even though they have never formally studied and implicated in sperm competition. Although all the Y-linked genes are great candidates to study in the context of sperm competition, there is no evidence that PRY (or WDY) contributes to sperm competition, only that they impairs sperm function when completely knocked out.

3. 1st page; last paragraph; 4th sentence: Probably informative to also point out that all of the y-linked fertility factors encode sperm proteins and that kl-2, 3 and 5 are amongst the most abundant proteins in sperm based on proteomic analyses. This is consistent with their presumed role in sperm function and more specifically as major axoneme structural components.

Reviewer #3 (Remarks to the Author):

Hafezi and colleagues used the CRISPR-Cas9 technique to knock out the WDY and PRY genes in *Drosophila melanogaster* to examine their function. WDY is a recognized gene from the Y chromosome of *D. melanogaster* and a strong candidate for the reproductive factor kl-1, which has previously been examined and described by classical genetics.

Their findings demonstrate that WDY knockout mutants have motile sperm (contrary to prior studies), but these sperm are unable to enter female spermatheca, explaining mutant male flies' sterility. They also discovered that the sperm tail beats in WDY knockouts are much diminished, which could explain why these sperm do not reach the spermatheca.

These results are convincing and clear, despite the lack of CRISPR mutants rescuing WDY to definitively prove the gene's function in sperm motility. I also understand that the position of the gene on the Y chromosome and its effects on sterility must make it difficult to obtain such recovered mutants.

Major comments

1) Section: Significant hydrophobicity differences in putative calcium-binding residues coincide with WDY's transition to Y-linkage in *Drosophila melanogaster*

The results show that the hydrophobicity of the WDY orthologs protein sequences from the *melanogaster* group (Y-linked gene) differs from the WDY from the mysterious group (A-linked). Yet, because *obscura* is a monophyletic group, it is impossible to determine whether the distinctions between *obscura* and *melanogaster* are ancestral or derivative. To find out, ortholog sequences from outgroup species must be included in this comparison.

Thankfully, given the abundance of *Drosophila* species genomes available in public sources, or even WDY sequences already annotated for other *Drosophila* species, this is a simple task (*D. mojavensis*,

D. virilis, *D. Grimshaw*, and *D. willistoni*, in all of them the WDY ortholog is in its ancestral autosomal site).

Although the association between increased hydrophobicity and transposition to Y is appealing, this idea lacks sufficient support, owing to the fact that no additional transposition to Y event has been detected in the WDY gene. WDY reverted to an autosome in the montium subgroup (melanogaster group), and the return to a less hydrophobic nature in such orthologs may be proof of this association (of course, the absence of evidence does not rule out the hypothesis)

I propose that the authors look for positive selection signatures in melanogaster WDY in comparison to autosomal orthologs from the montium subgroup, particularly in the EF hand motifs. That will strengthen their conclusions.

Minor comments

Page 3: substitute "We observed no obvious differences in the movement of WDY versus control sperm" to "Visual inspection show no obvious differences in the movement of WDY versus control sperm".

Page 4: "The lack of swimming WDY sperm in the uterus likely explains the inability of WDY sperm to enter the storage organs, though other defects may also exist and contribute (Qu et al., 2021)." Lack is not an adequate wording, as the data shows that is a small amount.

Page 5: "The number of stored PRY sperm remained low at 24 hASM (Figure S5), however, significantly more sperm entered storage organs if males and females were housed together overnight (Figure S5)". Please, include statistics.

Many thanks to all the reviewers for their thoughtful and helpful comments. Below is our response to each comment and an explanation of the changes we made to the manuscript in response. We feel the resulting manuscript is much improved.

Reviewer's Comments	Authors' Response
Reviewer #1 (Remarks to the Author): This manuscript focuses primarily on the functional role of a Y-linked gene, WDY, in the fertility of the D. melanogaster males. For that, the authors generate CRISPR mutants, finding evidence of decreased sperm motility and unusual distribution patterns in the female sperm storage organs relative to the wild type. The authors characterize different properties of the mutant sperm in order to narrow-down the effects on male fertility. Specifically, they find that WDY mutant sperm has low tail-beat frequency - compared to wildtype sperm- and that cannot swim in the uterus, preventing the sperm from entering into the spermathecas and the seminal receptacle. A similar, but more limited, analysis is done on the mutant for the PRY gene, which is also Y-linked. The results are also interesting, finding important differences with the conclusions related to the WDY mutant. Overall, mutant characterization is comprehensive and convincing. The results help us to better understand the genotype-phenotype link in the context of the functionality of Y-linked genes. The dissociation between tail beating and actual propulsion is an additional relevant observation.	Thank you. We are thrilled to hear that the reviewer found our paper interesting, comprehensive, and convincing.
Specific comments and concerns: 1. The section on the hydrophobicity differences in putative calcium-binding residues and the transition to Y-linkage in the Drosophila melanogaster species group is very different from the rest of paper, more speculative, and the	As the reviewer suggested, we revisited and updated the analysis from Singh et al 2014 in which positive selection was identified on WDY. We incorporated WDY sequences from additional species, ran the analysis on distinct clades to avoid synonymous saturation, and used

implications based on correlative patterns. It would be nice to see a better connection between their observations and previous results identifying the action of positive selection (Singh et al. 2014) or relative to new, more refined analyses that can identify particular codons as target of positive selection. Do those codons coincide with the loop residues of the Odd EF Hand motif?

alternate model-comparisons to identify any specific sites of selection within WDY. These results are presented in an updated Figure 4, three new supplementary tables, editing of the text for the corresponding section in 'Results', and analysis details in 'Methods'. The new sections are also appended at the end of this file.

Although we observed some signals of positive selection on WDY within the melanogaster clade in some tests, overall, our analysis did not support positive selection on WDY or its EF Hand domain. While WDY does not appear to undergo repeated substitutions, as we might have initially expected, our primary assertions still hold that (1) the change in hydrophobicity at position "X" in the EF Hand domain is likely to confer significant functional effects, and (2) this change occurred at an interesting transition point – when WDY became Y-linked.

We therefore adjusted our wording to reflect this in the abstract as follows.

~~Furthermore, we find significant changes in key residues of a putative calcium-binding domain of *WDY* orthologs, suggesting that *WDY*'s functional evolution coincides with its transition from autosomal to Y-linked in *Drosophila melanogaster* and its most closely related species.~~

Changed to...

Furthermore, we identify a significant change in hydrophobicity at a residue at a putative calcium-binding site in *WDY* orthologs at the split between the melanogaster and obscura species groups. This suggests that a major functional change in *WDY* coincided with its appearance on the Y chromosome.

	We omitted any reference to positive selection on WDY from the introduction. WDY and PRY are both evolving under positive selection (Singh et al., 2014) and Y-linked orthologs of WDY shows significant changes in key amino acid residues in a conserved calcium-binding domain, suggesting directed functional evolution of this gene. We also highlighted differences in the rate of evolution of the two EF Hand domain motifs rather than comparing between groups. While there was relative conservation of the domain within each group (82.4% identical sites in each group), there were notable differences between the groups (63.2% identical sites overall), particularly in the loop residues of the Odd EF Hand motif (Figure 4B). Changed to... We note a higher rate of sequence divergence of the Odd EF Hand Motif (55.9% Identity, Figure 4B) relative to that of the Even EF Hand motif (92.7% Identity, Figure 4C) in Drosophila sequences. We recognize that these assertions are correlative and speculative, as the reviewer points out, but we feel they are worth including for the benefit of future studies. We feel that in the absence of asserting positive selection, these hypotheses are now more specific and therefore stronger.
2. In the Conclusions section, I miss some discussion on the relationship between WDY and kl-1.	Thank you for the suggestion. We added the following sentences to the 'Conclusions' section: Our work extends Kiefer's intriguing observation that kl-1 mutants produce beating sperm despite being sterile. WDY mutants recapitulate this unusual phenotype. In the absence of the ability

	to test complementation, this strongly suggests that both mutants perturb the same gene.
3. 3. Please indicate in the main text the statistical test used next to the p-value.	Thank you for the suggestion. In our revision we made sure to specify each statistical test that we ran in both the main text and relevant figure legends.
Reviewer #2 (Remarks to the Author): Hafezi et al. have produced CRISPR KOs of two Y-linked genes in Drosophila and conducted a very thorough functional analysis of the resultant sperm and sperm storage phenotypes. The function and evolution of Y-linked genes has been of great interest to the field with regard to spermatogenesis and sexual selection but advances have been limited given the challenges of doing proper genetics involving the Y because KOs are going to result in male infertility. The authors have used a very clever crossing strategy using compound sex chromosomes to allow them to generate males with WDY and PRY KOs. So the combined use of CRISPR KOs and this mating strategy is very timely and provides the first real opportunity to carefully interrogate the functional role of Y-linked fertility factor. The analyses involved assay of sperm storage and in vitro sperm motility and were informative and robust. The manuscript is also very well written. As such, I only have a small number of substantive comments that need to be addressed. In summary, a very well-designed and informative study about how Y-linked genes contribute to sperm function and fertility.	Thank you, we really appreciate the reviewer's enthusiasm for the significance of this work, as well as their acknowledgement of the technical challenges that were involved.
Specific Comments: 1. Page 5; 3rd paragraph: WDY molecular evolution- Phylogeny almost certainly explains some, if not all, of this pattern between groups. A proper molecular evolutionary analysis is	We updated the phylogenetic selection analysis from Singh et al 2014 with additional sequences from four clades (for the whole protein) or five clades (for the EF Hand domain). This analysis

required to account for divergence between the melanogaster and obscura group.	(appended below) did not reveal evidence of positive selection. However, the available tests for positive selection rely on either consistently recurring amino acid changes at a single residue or an excess of changes throughout the whole molecule or a region of the molecule. We do feel that the hydrophobicity change in position “X” is likely functionally significant and worth pointing out. But, because melanogaster and obscura are so diverged, we cannot statistically identify this single instance of change as indicating selection. Future functional studies will be required to test the significance of this change, as we hypothesize.
Furthermore, isn't the expectation that WDY should evolve more rapidly once it moves to the Y? This could also be part of the model that is tested. Is the radical change at amino acid residue “X” a selected site? A formal analysis of WDY evolution should be conducted to better understand the selective dynamics associated with its relocation to the Y.	We do not necessarily expect to broadly see positive selection or rapid evolution of Y-linked genes. The haploid transmission of the Y might be expected to result in rapid response to new fitness-altering mutations. However, the Hill-Robinson effect means that the lack of recombination reduces the efficacy of selection acting on any specific variant. Each new mutation on the Y arises on a non-recombining haplotype, and the fitness of that haplotype is an average of all the variants on the haplotype. Only if the new mutation's effects outweigh the effects of other variants, as appears to be the case with WDY, will selection be able to drive it to fixation. The details of our formal analysis of WDY evolution, as requested, are included below.
2. Page 6; 2nd paragraph; last sentence: I'm a bit uncomfortable with this sentence even though the statements are qualified. First, no analyses of remating rates were conducted. It seems a bit premature to connect the dots here between reduced sperm storage numbers, SFP abundance	The reviewer raises an excellent point. This sentence has been removed.

in the FRT and remating rates. Furthermore, it also has the potential to be misleading to connect this to sperm competition, in turn. Several Drosophila genes now have “sperm competition” GO characterizations even though they have never formally studied and implicated in sperm competition. Although all the Y-linked genes are great candidates to study in the context of sperm competition, there is no evidence that PRY (or WDY) contributes to sperm competition, only that they impairs sperm function when completely knocked out.	
3. 1st page; last paragraph; 4th sentence: Probably informative to also point out that all of the y-linked fertility factors encode sperm proteins and that kl-2, 3 and 5 are amongst the most abundant proteins in sperm based on proteomic analyses. This is consistent with their presumed role in sperm function and more specifically as major axoneme structural components.	Thank you for the suggestion, we included it as follows: All of the Y-linked fertility factors encode sperm proteins; three of these, kl-2, 3, and 5, are among the most abundant proteins detected in sperm proteomic analyses (Garlovsky et al., 2022). This is consistent with their presumed role in sperm function and more specifically as major axoneme structural components (Carvalho et al., 2001, 2000; Goldstein et al., 1982).
Reviewer #3 (Remarks to the Author): Hafezi and colleagues used the CRISPR-Cas9 technique to knock out the WDY and PRY genes in Drosophila melanogaster to examine their function. WDY is a recognized gene from the Y chromosome of D. melanogaster and a strong candidate for the reproductive factor kl-1, which has previously been examined and described by classical genetics. Their findings demonstrate that WDY knockout mutants have motile sperm (contrary to prior studies), but these sperm are unable to enter female spermatheca, explaining mutant male flies' sterility. They also discovered that the sperm tail beats in WDY knockouts are much diminished, which could explain why these sperm do not reach the spermatheca.	We thank the reviewer for their positive assessment of our paper. Indeed, although a rescue would have been ideal, we decided not to attempt it because of the gene's extremely large size and the many complications involved in genetic manipulation of the Y chromosome.

These results are convincing and clear, despite the lack of CRISPR mutants rescuing WDY to definitively prove the gene's function in sperm motility. I also understand that the position of the gene on the Y chromosome and its effects on sterility must make it difficult to obtain such recovered mutants.	
Major comments 1) Section: Significant hydrophobicity differences in putative calcium-binding residues coincide with WDY's transition to Y-linkage in Drosophila melanogaster The results show that the hydrophobicity of the WDY orthologs protein sequences from the melanogaster group (Y-linked gene) differs from the WDY from the mysterious group (A-linked). Yet, because obscura is a monophyletic group, it is impossible to determine whether the distinctions between obscura and melanogaster are ancestral or derivative. To find out, ortholog sequences from outgroup species must be included in this comparison. Thankfully, given the abundance of Drosophila species genomes available in public sources, or even WDY sequences already annotated for other Drosophila species, this is a simple task (D. mojavensis, D. virilis, D. Grimshaw, and D. willistoni, in all of them the WDY ortholog is in its ancestral autosomal site).	Thank you for the suggestion. The sequences for these four species have now been added into the alignment in Figure 4. These sequences show the hydrophobic state to be ancestral.
Although the association between increased hydrophobicity and transposition to Y is appealing, this idea lacks sufficient support, owing to the fact that no additional transposition to Y event has been detected in the WDY gene. WDY reverted to an autosome in the montium subgroup (melanogaster group),	While most Y-linked genes (e.g. kl-2, ORY, etc.) were ancestrally Y-linked, WDY was ancestrally autosomal. The transition of WDY to the Y chromosome at the obscura-melanogaster split therefore represents the first instance when WDY became Y-linked in Drosophila. The significance of this event

and the return to a less hydrophobic nature in such orthologs may be proof of this association (of course, the absence of evidence does not rule out the hypothesis) I propose that the authors look for positive selection signatures in melanogaster WDY in comparison to autosomal orthologs from the montium subgroup, particularly in the EF hand motifs. That will strengthen their conclusions.	makes it less comparable to other instances of Y-autosome transition that we can examine. There are two additional transitions in our updated analysis in the montium species group, however, these appear to involve a whole-chromosome incorporation event as described in Dupim et al 2018 – affecting all of the Y-linked genes and likely involving a Y-autosome duplication event with one copy of WDY remaining on the Y-chromosome. We find that the residue of interest remains hydrophobic throughout the montium group, but this is difficult to interpret given the nature of the Y-incorporation event. We hypothesize that WDY functionally evolved coinciding with Y-linkage. As the reviewer notes, it does not necessarily follow that every instance of transition of WDY to the Y-chromosome involved the same exact change.
Minor comments Page 3: substitute “We observed no obvious differences in the movement of WDY versus control sperm” to “Visual inspection show no obvious differences in the movement of WDY versus control sperm”.	Thank you for the suggestion, this change was made, verbatim.
Page 4: "The lack of swimming WDY sperm in the uterus likely explains the inability of WDY sperm to enter the storage organs, though other defects may also exist and contribute (Qu et al., 2021)." Lack is not an adequate wording, as the data shows that is a small amount.	A great point - we reworded this sentence as follows: The inability of WDY sperm to enter the sperm storage organs likely reflects their diminished swimming in the uterus...
Page 5: “The number of stored PRY sperm remained low at 24 hASM (Figure S5), however, significantly more sperm entered storage organs if males and females were housed together overnight (Figure S5).”. Please, include statistics.	Thank you for the suggestion. We included more details on our experimental methods for characterizing the distribution of sperm in the female reproductive tract and we statistically analyzed the resulting data as follows:

	The distribution of sperm in the female reproductive tract was assessed from the images. Each region was scored as containing 0, less than or equal to 10, or more than 10 sperm. Distributions of sperm in the female reproductive tract were examined for 20-30 samples for each genotype. Differences in the distribution of sperm between different genotypes were compared using an Asymptotic Linear-by-Linear Association Test for an ordered categorical variable by genotype, stratified by reproductive tract region. The results of the statistical tests are now in both the main text and in the figure legends. The result for the specific section referred to by the reviewer was as follows: Indeed, the number of PRY sperm stored was significantly reduced compared to controls at 2 hASM (Figure 2F-H, p-value < 0.001, Asymptotic Linear-by-Linear Association Test for difference in sperm distribution). PRY mutant sperm were frequently absent or reduced in the seminal receptacle, and rarely observed in the spermathecae. The number of stored PRY sperm remained low at 24 hASM (Figure S5, p-value < 0.001, Asymptotic Linear-by-Linear Association Test for difference in sperm distribution), however, significantly more sperm entered storage organs if males and females were housed together overnight (Figure S5, p-value <0.001 for comparison of PRY versus control after overnight mating, <0.001 for comparison of PRY 24 hASM versus after overnight mating, and 0.9462 for comparison of control 24 hASM versus after overnight mating, Asymptotic Linear-by-Linear Association Tests for difference in sperm distributions).
--	---

New sections reporting details of selection analysis requested by the reviewers:

Edited 'Results' Section

Significant hydrophobicity differences in putative calcium-binding residues coincide with *WDY*'s transition to Y-linkage in the *melanogaster* lineage

Calcium regulates sperm motility in many organisms, including humans (Hong et al., 1984) and *Drosophila* (Gao et al., 2003; Köttgen et al., 2011; Watnick et al., 2003). *WDY*'s amino acid sequence contains a calcium-binding domain signature: an EF Hand (Interpro (Paysan-Lafosse et al., 2023)). Functional EF Hand domains contain a pair of motifs, each consisting of a loop flanked by alpha helices, that can bind Ca^{2+} ions. The specific characteristics of the loop affect calcium-binding affinity (Michiels et al., 2002). We identified a putative pseudo EF hand motif followed by a canonical EF hand motif in *WDY* (Methods, Figure 4A-C, Table S6). Known calcium-binding proteins (e.g. Calbindin D9K (Zhou et al., 2006)) also display this configuration. We also improved the annotation of two *WD40* domains (Methods, Figure 4A, Table S6), which typically mediate protein-protein interactions in protein complex assembly and/or signal transduction. Based on these findings, we speculate that *WDY* is necessary for sperm to recognize and adjust their motility based on differences in calcium in different regions of the RT.

We also compared sequences of the EF Hand domain between *WDY* orthologs in *Drosophila* species from three groups encompassing: (1) the initial transition of *WDY* to the Y chromosome from its ancestral autosomal site during the *melanogaster-obscura* group split, (2) a whole-chromosome Y-incorporation event in the *montium* subgroup, in which the Y chromosome is thought to have become duplicated elsewhere in the genome, and (3) the subsequent reestablishment of Y-linkage in the *kikkawai* clade (Figure 4B, C, (Dupim et al., 2018; Koerich et al., 2008)). We note a higher rate of sequence divergence of the Odd EF Hand Motif (55.9% Identity, Figure 4B) relative to that of the Even EF Hand motif (92.7% Identity, Figure 4C) in *Drosophila* sequences. The transition at position "X" is particularly compelling, since it involves a profound biochemical change in a conserved residue thought to directly bind calcium (Zhou et al., 2006). The shift away from canonical residues in the *melanogaster* group could indicate a modulation of calcium binding, and thus significant functional evolution in the EF Hand domain, coinciding with *WDY*'s initial Y-linkage. Corresponding shifts in hydrophobicity are not observed to be correlated with the genomic movements of *WDY* in the *montium* subgroup, but we would not necessarily expect a change that occurred when a gene moved to the Y chromosome to reverse if the gene moves off the Y chromosome. Moreover, selective pressures that drove a chromosome-wide Y-incorporation event are likely to have been significantly different from those driving movement of a single gene onto the Y chromosome. Future functional studies will be required to formally test the significance of the change in hydrophobicity at position "X".

A previous publication from our lab identified signatures of positive selection in *WDY*. We hypothesized that positively selected sites may be present within the EF hand domain. We thus updated the analysis of variation in *WDY* by including sequences from thirteen additional species (Table S7), a more stringent model comparison and statistical analysis to account for neutral evolution, and running the analysis on individual *Drosophila* clades, to avoid synonymous saturation. Our updated analysis (Table S8, S9) indicates that there is no evidence of positive selection that could not be better explained by neutral evolution. Furthermore, posterior probabilities for site classes, determined by Bayes Empirical Bayes ($p > 0.9$), identified no specific sites under positive selection. Therefore, although there seems to be an amino acid change in the EF hand domain as *WDY* became Y-linked, these sites cannot be shown to be undergoing positive selection. The initial movement of a gene to the Y chromosome may be more significant for its functional evolution than subsequent movements onto/off of the Y chromosome. This

is consistent with the observation of high incidence of gene re-appearance on the Y chromosome after Y-incorporation events (Dupim et al., 2018).

Updated Figure 4

Figure 4: Difference in hydrophobicity in the Odd EF Hand domain motif coincide with the initial movement of *WDY* to the Y chromosome at the melanogaster-obscura split

(A) Domain structure of *WDY*. Green indicates the two EF Hand motifs of the EF Hand domain. Browns indicate two WD40 domains made up of six and seven WD40 repeats. Repeat six of the first domain is present as two fragments. (B,C) Protein alignment of melanogaster and obscura group species for the region with the Odd (Pseudo) (B) and Even (Canonical) (C) EF Hand motif. Green bar indicates the motif, M indicates the position of any mismatch between the *melanogaster* sequence and the consensus, and black boxes indicate putative calcium-binding residues (X,Y,Z,-X,-Y,-Z). (D) Predicted AlphaFold structure of Odd EF Hand domain with putative calcium binding residues labelled, generated in PyMOL. Blue-red scale indicates hydrophobicity (Red is hydrophobic, Blue is hydrophilic).

New supplementary tables

Table S7: Source of sequences of WDY orthologues

Species	Source of DNA Sequence	Group				
		D.melanogaster	D.melanogaster , no D.triauraria	D.obscura	D.virilis	D.montium
D.melanogaster	NM_001316659.1 (full length)	•	•			
D.sechelia	Chang et al, 2022, eLife 11:e75795 (full length)	•	•			
D.simulans	Chang et al, 2022, eLife 11:e75795 (full length)	•	•			
D.mauritiana	Chang et al, 2022, eLife 11:e75795 (full length)	•	•			
D.erecta	Genbank: HQ852741.1 (missing N-terminus)	•	•			
D.yakuba	Genbank: BK006450.1 (missing N-terminus)	•	•			
D.asahinai	Extracted from VNJZ01001055.1 via Exonerate (EF-hand only)					•
D.lacteicornis	Extracted from VNKFO1012331.1 via Exonerate (EF-hand only)					•
D.tani	Extracted from VNJ001004106.1 via Exonerate (EF-hand only)					•
D.auraria	Extracted from VNJW01007069.1 via Exonerate (EF-hand only)					•
D.triauraria	Extracted from CM024333.1 via Exonerate (full length)	•				•
D.pectinifera	Extracted from VNKCO1003049.1 via Exonerate (EF-hand only)					•
D.anomelani	Extracted from JAEIIZ010011789.1 via Exonerate (EF-hand only)					•
D.ananassae	Genbank: EU362855.1 (full length)	•	•			
D.miranda	Extracted from NC_030306.1 via Exonerate (full length)			•		
D.persimilis	Extracted QMET02000011.1 from via Exonerate (full length)			•		
D.pseudoobscura	Genbank: BK006447.1 (full length)			•		
D.obscura	Extracted from JAEWW010000199.1 via Exonerate (full length)			•		
D.guanche	Extracted from OOUW01000006.1 via Exonerate (full length)			•		
D.subobscura	Extracted from CM017786.1 via Exonerate (full length)			•		
D.willistoni	Genbank: BK006446.1 (full length)			•		
D.virilis	Genbank: BK006445.1 (full length)				•	
D.novamexicana	Extracted from QMEP02000025 via Exonerate (full length)				•	
D.arizonae	Extracted from LSRM01000002.1 via Exonerate (full length)				•	
D.mojavensis	Genbank: BK006444.1 (full length)				•	
D.navojoa	Extracted from LSRL02000078.1 via Exonerate (full length)				•	
D.hydei	Extracted from QMEQ02000040.1 via Exonerate (full length)				•	
D.grimshawi	Genbank: BK006443.1 (full length)				•	

Table S8: Tests of Positive Selection on Full-Length WDY

We used PAML's codeml program to analyze nucleotide coding DNA sequences for signatures of positive selection. Model comparisons were performed between null models without positive selection and alternative models that include positive selection (M1a v M2a, M7 v M8, and M8a v M8). Model comparisons where the null is rejected are highlighted in gray. Although for the melanogaster clade, the null hypothesis was rejected for the M7 v M8 comparison, the null hypothesis was not rejected for the M8a v M8 comparison. The observed discrepancy in rejection of the null between both model comparisons likely indicates a false positive detection of selection by M7 v M8 comparison due to a subset of sites in WDY undergoing neutral evolution in the melanogaster clade. M7 restricts all categories of sites to have an omega <1, while M8a allows a subset of sites to have an omega = 1. Since M8a accounts for neutral evolution, it can be a more stringent null model for determining signatures of positive selection. Codeml analysis using all available WDY sequence does not show evidence of WDY undergoing positive selection. When analyzing all WDY sequences for positive selection, no selection was detected.

Group	Model Comparison	Null lnL	Alternative lnL	2x(Δ lnL)	df	p-value	omega	p(sites)	BEB sites with P>0.9
D. melanogaster	M1a v M2a	-10017.35696	-10017.35696	0	2	1			
	M7 v M8	-9989.681483	-9986.668286	6.03	2	0.049	1.18	0.01	
	M8a v M8	-9986.711538	-9986.668286	0.09	1	0.76			
D. melanogaster , no D. triauraria	M1a v M2a	-8507.453522	-8507.453522	0	2	1			
	M7 v M8	-8498.558319	-8496.304226	4.51	2	0.104			
	M8a v M8	-8496.664448	-8496.304226	0.72	1	0.4			
D. obscura	M1a v M2a	-9579.376445	-9579.376445	0	2	1			
	M7 v M8	-9563.499663	-9562.97558	1.05	2	0.59			
	M8a v M8	-9562.97558	-9562.97558	0	1	1			
D. virilis	M1a v M2a	-12026.53699	-12026.53699	0	2	1			
	M7 v M8	-12017.25419	-12015.6323	3.24	2	0.2			
	M8a v M8	-12015.67787	-12015.6323	0.09	1	0.76			
All Sequences	M1a v M2a	-24300.91375	-24300.91377	0	2	1			
	M7 v M8	-23949.18667	-23949.19039	0.01	2	1			
	M8a v M8	-23949.19039	-23949.19039	0	1	1			

Table S9: Tests of Positive Selection on EF Hand Domain of WDY

We restricted selection analysis to the EF hand domain and incorporated additional sequences from species in the montium subgroup for which full-length sequences were not available. We, again, used PAML's codeml program to analyze nucleotide coding DNA sequences for signatures of positive selection. Model comparisons were performed between null models without positive selection and alternative models that include positive selection (M1a v M2a, M7 v M8, and M8a v M8). Model comparisons where the null hypothesis was rejected are highlighted in gray. For the M7 v M8 comparison, the null hypothesis was rejected only for the *D. melanogaster* group when *D. triauraria* was included. However, the null hypothesis was rejected for the M8a v M8 comparison, indicating a likely false positive caused by neutral evolution.

Group	Model Comparison	Null lnL	Alternative lnL	2x(Δ lnL)	df	p-value	omega	p(sites)	BEB sites with P>0.9
D. melanogaster	M1a v M2a	-551.231303	-551.231306	0	2	1			
	M7 v M8	-557.146028	-551.401094	11.5	2	0.0032	1.7	0.015	
	M8a v M8	-551.654089	-551.401094	0.51	1	0.48			
D. melanogaster , no D. triauraria	M1a v M2a	-458.980644	-458.980645	0	2	1			
	M7 v M8	-459.812004	-459.107967	1.41	2	0.49			
	M8a v M8	-459.107993	-459.107967	0	1	1			
D. montium	M1a v M2a	-463.42136	-463.42136	0	2	1			
	M7 v M8	-463.072468	-463.072611	0	2	1			
	M8a v M8	-463.072611	-463.072611	0	1	1			
D. obscura	M1a v M2a	-740.774614	-740.774614	0	2	1			
	M7 v M8	-740.552849	-739.603289	1.9	2	0.37			
	M8a v M8	-740.13037	-739.603289	1.05	1	0.3			
D. virilis	M1a v M2a	-686.713995	-686.713995	0	2	1			
	M7 v M8	-686.079771	-686.080456	0	2	1			
	M8a v M8	-686.079866	-686.080456	0	1	1			
All Sequences	M1a v M2a	-1856.81276	-1856.81276	0	2	1			
	M7 v M8	-1809.417365	-1809.418055	0	2	1			
	M8a v M8	-1809.418031	-1809.418055	0	1	1			

Corresponding analysis details in 'Methods'.

Identification of WDY orthologue sequences

WDY ortholog sequences were obtained from GenBank, Chang et al 2022, or extracted from publicly available genomes using Exonerate version 2.2.0 (Slater and Birney, 2005), as noted in Supplementary Table S7. Newly extracted sequences were obtained by aligning the *D. melanogaster* WDY protein sequence (NM_001316659.1) to the genomic scaffolds containing WDY in other species via the Protein2Genome command. The top scoring prediction from Exonerate was used to define the sequence.

For the comparisons in Figure 4, protein sequences were aligned in Geneious software (RRID:SCR_010519) using a BLOSUM cost matrix with a gap open cost of 10 and a gap extend cost of 0.1.

Selection Analysis

To create sequence alignments for selection analysis, we translated the WDY coding sequences then aligned the protein sequences with MAFFT (Kato and Standley, 2013). Protein alignments were converted to nucleotide alignments by PAL2NAL (Suyama et al., 2006). Sites where >50% of the species had a gap in the alignment were removed from the final alignment used in the analysis. We pruned the phylogeny published in Suvorov et al 2022 to include only relevant species as input phylogenies for codeml analysis (Suvorov et al., 2022).

We used the codeml program of PAML 4.8 (Yang, 2007) to determine if there was evidence of positive selection in WDY and to potentially identify specific codons subject to positive selection. We compared neutral models to models including positive selection (M1a v. M2a, M7 v. M8, and M8a v. M8) via a likelihood ratio test (LRT). The LRT statistic was calculated from the model likelihoods as $2 \times (\text{Alternative} - \text{Null})$. For M1a v M2a and M7 v M8 comparisons, the LRT statistic was compared to the Chi-squared distribution with 2 degrees of freedom (Yang, 2007). For the M8a v M8 comparison, 1 degree of freedom was used (Swanson et al., 2003). Specific codons evolving under positive selection were identified via M8 in codeml with Bayes Empirical Bayes probabilities > 0.9 (Swanson et al., 2003).

Reviewers' comments:

Reviewer #1 (Remarks to the Author):

I appreciate the effort made by the authors to address my concerns.

Reviewer #2 (Remarks to the Author):

I've reviewed the responses to the suggestions from all three reviewers and feel that the authors have addressed them in a reasonable and thorough manner. It's a very nice study. Congratulations!

Reviewer #4 (Remarks to the Author):

It has been long known that the Y-chromosome of *D. melanogaster* contains six fertility factor and that the *kl-1* locus mutants produce mature sperm despite their sterility. Here the authors provide a functional characterization of a *kl-1* locus gene, *WDY*, by using a CRISPR knockout. The mutants produce mature sperm that accumulates in the posterior end of the testes causing that region to bulge. The sperm is motile and present, although in lower numbers, in the seminal vesicle. The *WDY* mutant sperm is transferred to females but has reduced motility in uterus, which might cause them to fail to enter the female storage organs. A similar effect on sperm storage was found for a *PRY* CRISPR KO, which showed a significant reduction of sperm stored compared to controls. The work is very thorough, and the results provide a needed detailed molecular functional characterization of the role of these genes in male fertility.

The authors also improve the annotation of the *WDY* gene and identify higher amino acid divergence between species in the odd EF-hand domain, with a significant amino acid change. This single amino acid change modifies hydrophobicity and could mean a major functional change coinciding with Y-linkage in the *melanogaster* group.

Overall, I find the writing and presentation of results very clear. The detailed functional characterization of the mutants is a very welcomed and much needed addition to an otherwise lacking functional characterization of genes in general and Y-linked fertility factors in particular. The work also raises some interesting questions about the evolution of these genes.

The only weak aspect for me is the evolutionary genetics analysis. It is not surprising that the tests of selection using the PAML codon models find no evidence of positive selection because the codon tests you conducted will need persistent selection over the phylogenetic scale for the signal to be detected. I don't think the test you used is the most appropriate. I am wondering why you did not include a simpler comparison of dN/dS ratio between the Odd EF hand, the even EF hand and the rest of the protein. The amino acid divergence is faster at the odd hand, but does it show a dN/dS ratio expected for a region under positive selection? Also, given an a priori hypothesis of a role of localization on the Y-chromosome on functional diversification, why not conducting a branch or branch-site model test to see whether signals of selection are detectable in the *melanogaster* - *obscura* split?

Reviewer #4 Comments:

The only weak aspect for me is the evolutionary genetics analysis. It is not surprising that the tests of selection using the PAML codon models find no evidence of positive selection because the codon tests you conducted will need persistent selection over the phylogenetic scale for the signal to be detected. I don't think the test you used is the most appropriate. I am wondering why you did not include a simpler comparison of dN/dS ratio between the Odd EF hand, the even EF hand and the rest of the protein. The amino acid divergence is faster at the odd hand, but does it show a dN/dS ratio expected for a region under positive selection? Also, given an a priori hypothesis of a role of localization on the Y-chromosome on functional diversification, why not conducting a branch or branch-site model test to see whether signals of selection are detectable in the melanogaster - obscura split?

Response to Reviewer:

We appreciate the additional helpful feedback from Reviewer #4 on the molecular evolutionary analysis. The reviewer is correct that the codon models of selection require persistent selection over the phylogenetic scale. We tried to mitigate this issue in our prior analysis by applying the codon models to individual clades of *Drosophila* rather than across all *Drosophila*. However, we also took the reviewer's suggestion to run additional tests to bolster our analysis.

We first performed a comparison of the dN/dS ratio (omega) between the Odd EF Hand, the Even EF Hand, and the full protein. We used model 0 in codeml to obtain the dN/dS values and performed the test on a pairwise comparison between *D. melanogaster* and *D. obscura* (outgroup sequence pre-translocation of WDY to Y chromosome) and using all available WDY sequences. We observed an estimated dN/dS value far below 1 in all cases, indicative of the entire WDY molecule evolving under strong purifying selection. This result is consistent with our codon model analysis. We also observed that the dN/dS for the Odd EF hand was higher than that of the Even EF Hand (~12 or 6 fold higher in the pairwise or all-sequence comparisons respectively). This result is consistent with our sequence identity comparison, however, both are qualitative comparisons, so it remains unclear how significant the difference is because the models are not nested, precluding a hypothesis test, and the omega values for all three are quite low.

Model 0	Odd dN/dS (omega)	Even dN/dS (omega)	Full Length dN/dS (omega)
Pairwise comparison between D.melanogaster and D.obscura	0.00239	0.00020	0.0154
All sequences comparison	0.09200	0.01576	0.05226

We added the result from the pairwise comparison into the paper as follows:

(Results section)

Similarly, we observed a 12-fold difference in d_N/d_S between the Odd EF hand (omega = 0.00239) and the Even EF hand (omega = 0.00020) in a pairwise comparison between *D. melanogaster* and *D. obscura* WDY using Model 0 of codeml. This supports the idea that the Odd EF hand sequence is diverging more rapidly or has more relaxed purifying selection.

(Methods section)

We used the codeml program of PAML 4.8 (Yang, 2007) to determine if there was evidence of positive selection in WDY and to potentially identify specific codons subject to positive selection. We compared neutral models to models including positive selection (M1a v. M2a, M7 v. M8, M8a v. M8) via a likelihood ratio test (LRT). The LRT statistic was calculated from the model likelihoods as $2 * [\log(L_a) - \log(L_0)]$, where L_a and L_0 are the likelihoods under the alternate and null hypotheses respectively. For M1a v M2a and M7 v M8 comparisons, the LRT statistic was compared to the Chi-squared distribution with 2 degrees of freedom (Yang, 2007). For the M8a v M8 comparison, 1 degree of freedom was used (Swanson et al., 2003). Specific codons evolving under positive selection were identified via M8 in codeml with Bayes Empirical Bayes probabilities > 0.9 (Swanson et al., 2003). We also used model 0 in codeml to obtain \$d_N/d_S\$ values on a pairwise comparison between *D. melanogaster* and *D. obscura* and performed a branch-site model test (M0 v. M2) using all available WDY sequences.

We next conducted a branch-site model test (Model 0 v. Model 2 comparison - not to be confused with the NSites models used for the sites tests for selection in certain clades) using all available full length WDY sequences to determine if selection on the branch where WDY moved to the Y-chromosome was indicative of positive selection. For this test, the alternative model (model 2) poses that all branches have the same estimated omega, except the specified branch where the autosome-to-Y translocation occurred. This is compared to the null model where one d_N/d_S value is estimated for all branches. In this test, the null model was rejected (p-value < 0.00001), however, the estimated omega value for this branch was indicative of purifying selection (omega < 1), not positive selection. Therefore, the branch-site model test was, again, consistent with our codon models of selection comparison across *Drosophila* and for individual clades in rejecting positive selection. The results from this test were added as Supplemental Table S10.

(Results section)

A previous publication from our lab identified signatures of positive selection in WDY. We hypothesized that positively selected sites may be present within the EF hand domain. We thus updated the analysis of variation in WDY by including sequences from thirteen additional species (Table S7), a more stringent model comparison and statistical analysis to account for neutral evolution, and running the analysis on individual *Drosophila* clades, to avoid synonymous saturation. Our updated analysis (Table S8, S9) indicates that there is no evidence of positive selection that could not be better explained by neutral evolution. Furthermore, posterior probabilities for site classes, determined by Bayes Empirical Bayes (p > 0.9), identified no specific sites under positive selection. Similar results were obtained with a branch-site test on full-length WDY (Table S10). Therefore, although there seems to be an amino acid change in the EF hand domain as WDY became Y-linked, these sites cannot be shown to be undergoing positive selection. The initial movement of a gene to the Y chromosome may be more significant for its functional evolution than subsequent movements onto/off of the Y chromosome. This is consistent with the observation of high incidence of gene re-appearance on the Y chromosome after Y-incorporation events (Dupim et al., 2018).

Table S10: Branch-site Test of Positive Selection on Full-Length WDY

We used PAML's codeml program to run a branch-site test for positive selection. Model comparisons were performed between null model (M0) without positive selection and alternative model (M2) that include positive selection on the branch where WDY moved to the Y-chromosome. Although the model passes, positive selection was not detected on the branch leading to species where WDY is Y-linked.

Results of Branch-site Test on All Sequences	
Model Comparison	M0 v M2
Null Model	-24534.935013
Alternative Model	-24521.014917
Chi-Square	27.84
df	1
p-value	<.00001
dN/dS - Y-linked branch	0.0136
dN/dS - autosomal branch	0.056

(Methods section)

We used the codeml program of PAML 4.8 (Yang, 2007) to determine if there was evidence of positive selection in WDY and to potentially identify specific codons subject to positive selection. We compared neutral models to models including positive selection (M1a v. M2a, M7 v. M8, M8a v. M8) via a likelihood ratio test (LRT). The LRT statistic was calculated from the model likelihoods as $2 * [\log(L_a) - \log(L_0)]$, where L_a and L_0 are the likelihoods under the alternate and null hypotheses respectively. For M1a v M2a and M7 v M8 comparisons, the LRT statistic was compared to the Chi-squared distribution with 2 degrees of freedom (Yang, 2007). For the M8a v M8 comparison, 1 degree of freedom was used (Swanson et al., 2003). Specific codons evolving under positive selection were identified via M8 in codeml with Bayes Empirical Bayes probabilities > 0.9 (Swanson et al., 2003). We also used model 0 in codeml to obtain d_N/d_S values on a pairwise comparison between *D. melanogaster* and *D. obscura* and performed a branch-site model test (M0 v. M2) using all available WDY sequences.

Finally, we performed a branch model test which estimates different omegas for each branch of the phylogeny and compares it to the same null model used for the branch-site mode. Again, we did not detect any evidence of positive selection coinciding with the translocation of WDY to being Y-linked. Although the branch model passed, we did not detect any branches with an omega >1 in the clade containing the translocation to Y-linked by WDY. Some other branches had very high omegas, but these were likely spurious due to them being short branches with relatively few mutations (e.g. A branch with a dN of 0.015696 and a dS of 0.000095 would have an omega value of > 165). The branch-site model is the better model for our study since it directly tests a hypothesis regarding evidence of selection along a specific branch. The branch model is overparameterized, making rejections of the null model rare, so we opted not to include it in the manuscript. However, we included the results of the branch model below in case the reviewer is interested.

Results of Branch Model Test on All Sequences	
Null Model	-24534.935013
Alternative Model	-24364.754584
Chi-Square	340.36
df	41
p-value	<.00001

REVIEWERS' COMMENTS:

Reviewer #4 (Remarks to the Author):

I would like to thank the authors for carefully addressing all my questions, and for incorporating results from the latest analysis when appropriate.

Reviewer #5 (Remarks to the Author):

The authors used CRISPR knockout of the Y-linked *Drosophila melanogaster* gene *WDY* to show that it is necessary for sperm motility and male fertility. This is primarily a functional study and I think the authors have done a very good job both experimentally and in presenting their results. I was not a reviewer of the original submission, but am commenting on the revised manuscript, especially the point raised by the original Reviewer 4. Here I think the authors have satisfactorily addressed the Reviewer comment. They used state-of-the-art methods to test for positive selection, employing various models for testing the dN/dS ratio across the *Drosophila* phylogeny. In the end, they didn't find convincing evidence for positive selection, either in particular functional domains or on the branch leading to Y-linkage. This is not too surprising, because properties of the Y (no recombination, reduced effective population size) make it very unlikely that there would be repeated rounds of positive selection acting on this gene, which would be required to see a proportional increase in non-synonymous substitutions. As the authors suggest in the paper, it maybe be that the re-location to the Y itself (or a single amino acid replacement accompanying this re-location) was the main selective event. I think they are right to be cautious about making conclusions about selection, particularly with the final sentences of the conclusions section.